# Clustered Regularly Interspaced Short Palindromic Repeat Analysis of Clonal Complex 17 Serotype III Group B *Streptococcus* Strains Causing Neonatal Invasive Diseases

**DOI:** 10.3390/ijms222111626

**Published:** 2021-10-27

**Authors:** Jen-Fu Hsu, Jang-Jih Lu, Chih Lin, Shih-Ming Chu, Lee-Chung Lin, Mei-Yin Lai, Hsuan-Rong Huang, Ming-Chou Chiang, Ming-Horng Tsai

**Affiliations:** 1Division of Pediatric Neonatology, Department of Pediatrics, Chang Gung Memorial Hospital, Taoyuan 333, Taiwan; jeff0724@gmail.com (J.-F.H.); annielin85@gmail.com (C.L.); kz6479@cgmh.org.tw (S.-M.C.); lmi818@msn.com (M.-Y.L.); qbonbon@gmail.com (H.-R.H.); cmc123@cgmh.org.tw (M.-C.C.); 2School of Medicine, College of Medicine, Chang Gung University, Taoyuan 333, Taiwan; 3Department of Laboratory Medicine, Chang Gung Memorial Hospital at Linkou, Taoyuan 333, Taiwan; leollc@gmail.com; 4Department of Medical Biotechnology and Laboratory Science, Chang Gung University, Taoyuan 333, Taiwan; 5Division of Neonatology and Pediatric Hematology/Oncology, Department of Pediatrics, Chang Gung Memorial Hospital, Yunlin 638, Taiwan

**Keywords:** group B *Streptococcus*, CRISPR, multilocus sequence typing, antimicrobial resistance, phage

## Abstract

Group B *Streptococcus* (GBS) is an important pathogen of neonatal infections, and the clonal complex (CC)-17/serotype III GBS strain has emerged as the dominant strain. The clinical manifestations of CC17/III GBS sepsis may vary greatly but have not been well-investigated. A total of 103 CC17/III GBS isolates that caused neonatal invasive diseases were studied using a new approach based on clustered regularly interspaced short palindromic repeats (CRISPR) loci and restriction fragment length polymorphism (RFLP) analyses. All spacers of CRISPR loci were sequenced and analyzed with the clinical presentations. After CRISPR-RFLP analyses, a total of 11 different patterns were observed among the 103 CRISPR-positive GBS isolates. GBS isolates with the same RFLP patterns were found to have highly comparable spacer contents. Comparative sequence analysis of the CRISPR1 spacer content revealed that it is highly diverse and consistent with the dynamics of this system. A total of 29 of 43 (67.4%) spacers displayed homology to reported phage and plasmid DNA sequences. In addition, all CC17/III GBS isolates could be categorized into three subgroups based on the CRISPR-RFLP patterns and eBURST analysis. The CC17/III GBS isolates with a specific CRISPR-RFLP pattern were more significantly associated with occurrences of severe sepsis (57.1% vs. 29.3%, *p* = 0.012) and meningitis (50.0% vs. 20.8%, *p* = 0.009) than GBS isolates with RFLP lengths between 1000 and 1300 bp. Whole-genome sequencing was also performed to verify the differences between CC17/III GBS isolates with different CRISPR-RFLP patterns. We concluded that the CRISPR-RFLP analysis is potentially applicable to categorizing CC17/III GBS isolates, and a specific CRISPR-RFLP pattern could be used as a new biomarker to predict meningitis and illness severity after further verification.

## 1. Introduction

*Streptococcus agalactiae* (Group B *Streptococcus*; GBS) is part of the commensal flora in the human genitourinary and gastrointestinal tracts, but GBS is one of the most important pathogens that cause neonatal sepsis and meningitis and sometimes adult invasive diseases [1,2]. The colonization rate of GBS in pregnant women is approximately 18–35%, the infection rate in neonates is approximately 0.4 to 1.1 cases per 1000 live births, and the GBS-sepsis-attributable mortality rate is approximately 10–18% [3,4,5]. The molecular and epidemiological information on GBS isolates in colonized women and their children has been the focus of numerous studies [6,7,8] because these data are especially important for the prevention of neonatal GBS sepsis and the development of effective GBS vaccination [9,10,11].

The differences in capsular polysaccharides (CPS) define GBS isolates into ten different serotypes (Ia, Ib, and II to IX), with serotype III emerging as the most common strain that causes neonatal sepsis in most countries [5,11,12]. In addition, multilocus sequence typing (MLST) has been developed to optimize the discrimination of specific GBS strains to study the epidemiology of GBS infection [11,12,13,14]. The sequence types (STs) are identified by MLST using a combination of alleles for seven housekeeping genes, and genetically related STs could be clustered into clonal complexes (CCs) following phylogenetic analyses. This subtyping system consists of CPS, and STs are now applied by the majority of current GBS studies [11,12,13,14].

Clustered regularly interspaced short palindromic repeats (CRISPR) are composed of short (25–40 base pairs) direct repeats interspaced by nonrepetitive similar-sized sequences called spacers [15,16]. CRISPR-associated protein-9 (Cas9)-mediated genome modification is a novel RNA-domain-containing endonuclease-based genome editing technology for a variety of biological and therapeutic applications [17,18,19]. CRISPR/Cas9-based genomic editing of various bacterial strains, such as *Escherichia coli*, *Clostridia,* and *Streptomycetes*, has been performed to investigate the effects of gene deletion, insertion, replacement, and gene regulation [19,20]. CRISPR loci are characterized by a dynamic, rapidly evolving, and polymorphic composition because the system is able to capture novel spacers derived from mobile genetic elements (MGEs) in the environment [18,19,20]. Therefore, spacer polymorphism and acquisition of novel spacers of CRISPR loci can be epidemiological markers for transmission route identification and bacterial genotyping. Recent studies have also documented that CRISPR analysis is an applicable tool to follow maternal GBS colonization and separate GBS isolates with greater discriminating power than traditional subtyping systems [20,21,22]. The CRISPR loci of different GBS strains are highly diverse and reveal the dynamic nature of this system [20,21,22]. However, the characteristics of the CRISPR loci of the same GBS strains have not yet been investigated. In this study, we aimed to investigate the homogeneity of CC17/III GBS strains from neonatal invasive diseases.

## 2. Results

From January 2006 to December 2018, 103 nonduplicate CC17/III GBS isolates from 103 neonates with invasive GBS disease in CGMH were collected for analyses. There were a total of 8 patients with early-onset disease and 95 patients with late-onset disease. The median (interquartile range (IQR)) gestational age and birth weight of this cohort were 39.0 (37.0–40.0) weeks and 2940 (2655.0–3360.0) g, respectively. Only 19 (18.4%) were premature neonates (GA < 37 weeks), and only 4 neonates were very low birth weight infants (BBW < 1500 g). The onset of GBS sepsis occurred at 29.0 (15.0–51.0) (median (IQR)) days of life. There were four cases of very late GBS sepsis (disease onset > 90 days old). Among these patients, 29 (28.1%) had meningitis. MLST analysis and CPS serotyping were performed to confirm that all these GBS isolates were ST17/III GBS strains.

### 2.1. CRISPR-RFLP and CRISPR Direct Repeat Analysis

For each isolate, we performed CRISPR-RFLP with the restriction enzyme DdeI. The resulting restriction fragments were separated according to length by gel electrophoresis (Figure 1). After RFLP and electrophoresis, all PCR results of CRISPR1 from these GBS isolates showed three bands, with lengths of CRISPR1 sequences ranging from 140 to 1250 bp. The common 140 and 270 bp fragments were part of the type II-A CRISPR-associated protein and hypothetical protein, respectively. Among all GBS isolates, a total of 11 different gel band patterns were detected by CRISPR-RFLP (Table 1). Because previous studies showed that GBS isolates with the same RFLP pattern may contain similar CRISPR spacer compositions [21], the CRISPR compositions of all isolates were sequenced and analyzed to document this phenomenon. In addition, the PCR products of CRISPR-RFLP were sequenced, and we found that GBS isolates with the same RFLP pattern also had similar spacer compositions (Table 1). Most of the spacers were identified in previous studies, and the same represented numbers were coded [23]. We found that 82 out of 103 (79.6%) GBS isolates contained seven spacers in a conserved order (101, 102, 49, 53, 54, 55, and 56). A total of 18 new spacers were found in this cohort, submitted to GeneBank, and assigned as S1 to S18. We found that all of the newly identified spacers were in GBS strains with larger-molecular-weight CRISPR-RFLP bands. In addition, it is well-known that new spacers are inserted at the 5′ end of the array in most cases, and we suggest that these spacers may have been acquired recently after verification by checking the positions of the spacers.

Among the 103 CC17/III GBS isolates, a total of 966 repeats were identified, including a highly conserved typical repeat presenting in all isolates and representing 89.3% of the 966 repeats identified, and 100% of the internal repeats. BLASTn and CRISPRdb were used to analyze the typical repeat sequence and investigate whether it had homology to other bacterial sequences. We found that the typical repeat sequence (5′-GTTTTAGAGCTGTGCTGTTTCGAATGGTTCCAAAAC-3′) and terminal repeat sequence (5′-GTTTTAGAGCTGTGCTGTTATTATGCTAGGACATCA-3′) were identical to those found by Lier et al. [24]. Therefore, this typical repeat showed sequence similarity to other typical repeats found in species of the Streptococcus genus [25].

### 2.2. CRISPR1-Based Subgroups of CC17/III GBS Isolates and Spacer Analysis

The phylogenetic analysis of 103 GBS isolates was carried out using eBURST software based on the diversity and relatedness of CRISPR type, as shown in Figure 2. All 103 GBS strains were divided into three phylogenetically distinct groups. The patient demographics and clinical presentations of the three subgroups are presented in Table 2. We found that the patient demographics were not significantly different between the three subgroups. However, neonatal sepsis caused by GBS isolates in group A had significantly higher rates of severe sepsis (63.6% vs. 30.9%, *p* = 0.002) and meningitis (50.0% vs. 22.2%, *p* = 0.015) than GBS isolates in groups B and C. In addition, the CRISPR-RFLP band lengths of GBS isolates in group A were mostly <1000 bp or >1300 bp, whereas those in group B and C were between 1000 and 1300 bp. Therefore, we found that the CC17/III GBS isolates with RFLP lengths ≥1300 bp and <1000 bp were more significantly associated with the occurrence of severe sepsis (57.1% vs. 29.3%, *p* = 0.012) and meningitis (50.0% vs. 20.8%, *p* = 0.009) than GBS isolates with RFLP lengths between 1000 and 1300 bp. The overall mortality of this cohort was 3.9% (a total of four patients died).

All 103 GBS strains were sensitive to penicillin, ampicillin, and vancomycin. None of these strains displayed cefotaxime resistance. On the other hand, only 8.7% (*n* = 9) and 14.6% (*n* = 15) of all GBS strains were susceptible to erythromycin and clindamycin, respectively. Most of the erythromycin-resistant GBS isolates (51.1%) were also not susceptible to clindamycin (94.6%, 87 isolates, *p* < 0.001 by Pearson’s χ^2^ test). In addition, all the GBS isolates in group C were antibiotic resistant, which was relatively higher than other groups.

BLASTn analyses were performed to analyze the homologous sequences of all the spacers to determine their potential source and function (Table 3). The CRISPR spacers are generally segments of DNA captured from invasive plasmids or phages. We found that more than 86% of these spacers had similar sequences in the NCBI database, with 65% (*n* = 28) of these spacers showing homology to phage sequences, 18.6% (*n* = 8) to the GBS chromosome, and only one spacer to a plasmid sequence. Only one spacer (2.3%) showed homology to plasmid, which indicated that horizontal gene transfer by plasmids was uncommon in CC17/III GBS strains. In addition, seven spacers showed homology to the same phage LF2, five spacers showed homology to the same phage Javan52, and three spacers showed homology to the same phage Javan48. Among those with matched MGEs, the majority of the spacers (24/29, 82.8%) displayed 100% homology to the corresponding sequences in phages or plasmids, and only five spacers had between one and seven base mutations. The correspondences between spacers and patients’ clinical conditions are presented in Appendix A and no specific conclusions can be documented.

### 2.3. WGS Analyses of CC17/III GBS Isolates with Different CRISPR-RFLP Patterns

WGS was conducted in CC17/III GBS isolates with different CRISPR-RFLP patterns in order to verify the results (Appendix A). A total of eighteen CC17/III GBS isolates, including six in group A, six in group B, and six in group C, were chosen to investigate the genetic differences that may potentially cause severe sepsis and meningitis. We found that the genes of component systems CovS/R, capsular serotypes, presence of pili island genes, and most virulence genes were not significantly different between all these CC17/III GBS isolates. The only differences were the antibiotic-resistance genes. We found that the significant difference was the presence of one integrative and conjugative element, ICE*Sag37*, carrying multiple antibiotic resistant genes and virulence genes, which was found in antimicrobial-resistant CC17/III GBS isolates only and absent in antimicrobial-susceptible CC17/III GBS isolates. The presence of ICE*Sag37* was not significantly different between different CRISPR-RFLP patterns and between the three subgroups. Additionally, the presence of ICE*Sag37* was not associated with the occurrence of severe sepsis and meningitis [6].

## 3. Discussion

In this study, we investigated the molecular characteristics of invasive GBS strains by comparing the CRISPR1 loci of 103 CC17/III GBS isolates from 103 different neonates over an 11 year period. We aimed to assess the diversity of spacer content of the same GBS serotype over time and evaluate whether the CRISPR1 locus can be an epidemiological marker, since it can reflect the evidence of phage infection in vivo and is potentially associated with biological functions [17,18,19,20]. Based on CRISPR-RFLP analysis and CRISPR1 locus diversity, we can divide the invasive CC17/III GBS strains into several categories because GBS isolates with the same CRISPR-RFLP length have nearly 100% similarity in spacer compositions. Based on different CRISPR subtypes, certain specific subtypes of CC17/III GBS isolates were significantly associated with the occurrence of bacterial meningitis, severe sepsis, and septic shock.

Previous studies have demonstrated that the approach of using CRISPR-RFLP analysis to compare the CRISPR compositions, followed by sequencing and CRISPR array analyses, can be useful in clinical applications of screening GBS isolates and following maternal or neonatal carriage [20,21,22,25]. In these studies, CRISPR analyses were performed in a large number of GBS isolates of different serotypes and sequence types from different subjects, and there was a strong agreement between traditional MLST and CRISPR genotyping [20,21,22,24]. In addition, distinctive features of CRISPR loci were identified between different clonal complexes that are representative of different GBS strains [22,25]. It is well-known that the CC17/III GBS strain has emerged as the major pathogen causing neonatal invasive diseases and meningitis because this hypervirulent strain has specific virulence factors and adhesion molecules that can assist in invasion [26,27,28]. However, the clinical manifestations of all CC17/III GBS strains may vary greatly at presentation, as some of them present with fulminant sepsis and meningitis. These large variations have not been investigated, and this is the first study to use the CRISPR array and spacer compositions to subgroup these invasive CC17/III GBS isolates. In contrast to a previous study that found less spacer variation within the CC17 isolates [22,24], we documented that the diversity within the CC17 isolates was much higher, which can be explained partly by the significantly larger number of CC17 isolates in this study.

Investigation into the likelihood of phage infection can be performed in vivo through alterations in the spacer content and addition at the leader site of CRISPR1 [17,18,19,25]. The spacers of CRISPR are obtained from foreign DNA and can protect bacteria from attack by homologous phages or plasmids in the environment. Previous studies compared interspecies CRISPR loci and documented that the number of spacers is inversely correlated with prophage acquisition [29]. Because GBS is a significant bacterial pathogen of neonates and immunocompromised adults, continual genome evolution may result in the emergence of new virulent strains [29,30,31]. In our cohort, identical CRISPR1 loci among CC17 strains from different subjects were observed, which contrasts with previous studies that suggested the spacer content of the CRISPR1 locus is generally specific for isolates from a given subject [21,22]. In addition, this longitudinal study was performed using GBS strains over a long period of 11 years. Therefore, our study demonstrated the predominance of certain clones to emerge as the major strain causing neonatal invasive diseases in our institute, which is consistent with a recent study that found that a few clones comprised the majority of the invasive GBS isolates [32]. Further studies that enroll more CC17 GBS strains from multiple centers in Taiwan are warranted to document this phenomenon.

It is known that spacer acquisition at the 5′ leader-end plays an important role for CRISPR array diversity, but in vitro spacer acquisition has never been found in CC17/III GBS isolates [22,24]. Most of our CC17/III GBS isolates (95.1%, *n* = 98) had spacer compositions of 49-53-54-55-56, and we found that 81.6% (*n* = 80) of them had spacer acquisition, whereas 11.7% (*n* = 12) had spacer loss. Our data support the recent findings of Pastuszka et al. that spacer acquisition, sometimes accompanied with the loss of old spacers, occurs and may be more than ~10% of the ST17 clones [33]. Currently, only very few studies are available regarding the CC17/III GBS lineage [34], and this phenomenon requires further confirmation after enrolling a larger number of strains.

To the best of our knowledge, this is the first study to find a significant relationship between the CRISPR-RFLP pattern and clinical manifestations of GBS invasive diseases. Interestingly, GBS isolates with RFLP lengths between 1000 and 1300 bp were associated with a lower likelihood of severe sepsis and meningitis. In contrast, GBS isolates with RFLP length ≥1300 bp and ≤1000 bp were significantly more likely to cause severe illness, and GBS isolates within the same subgroup based on the RFLP length were always genetically related to each other. In addition, GBS isolates with the same CRISPR-RFLP pattern were found to have similar CRISPR loci and spacer sequences, as documented in this study and others [21]. Various virulence factors, including HvgA, pilus, and Srr1/Srr2, have been associated with meningitis through enhanced adhesion to epithelial cells and penetration of the blood–brain barrier [27,35,36], but the differences in these gene sequences were not associated with higher illness or meningitis in our cohort. We hypothesize that the CRISPR/Cas system in CC17/III GBS isolates can affect at least some biological functions that may contribute to the occurrence of meningitis. In addition, the CRISPR-RFLP assay may potentially be used as a new biomarker to predict clinical meningitis after further verification.

In conclusion, the diversity of the CRISPR1 sequences among CC17/III GBS isolates was demonstrated in this study, which represents a powerful and applicable subgrouping method for better investigation of specific clone dissemination. The CRISPR diversity is much higher even within a given GBS clonal complex [23], and involves spacer gain, random loss, and occasional duplication. Within the CC17/III GBS strains, a specific sublineage, which is identifiable by the ubiquitous CRISPR1 locus and spacer organization, is potentially associated with a higher severity of illness and a higher likelihood of meningitis. Further studies that enroll more clinical CC17/III GBS isolates from multiple centers are needed to verify these findings, since it is applicable to use the CRISPR-RFLP assay to identify the most dangerous CC17/III GBS strain in clinical practice and prompt more aggressive therapeutic strategies in advance.

## 4. Materials and Methods

### 4.1. Bacterial Isolates

From January 2008 to December 2018, all GBS strains isolated from blood and/or cerebrospinal fluid of neonates with invasive GBS diseases were obtained from the bacterial library of Chang Gung Memorial Hospital (CGMH, Taoyuan, Taiwan). All identifications of initial GBS isolates and storage in the bacterial library were based on the standard procedures: MALDI-TOF MS (Bruker corporation, Karlsruhe, Germany) was used for documentation of all GBS isolates (https://www.ncbi.nlm.nih.gov/pmc/articles/PMC8328194/, accessed on 24 February 2021). All GBS strains were genotyped using MLST and CPS serotyping in our previous studies [37,38]. Among them, CC17/III GBS isolates were retrieved to analyze their CRISPR loci. The clinical features of all neonates with CC17/III GBS invasive diseases were collected and followed [37,38]. This study was approved by the ethics committee of CGMH, and a waiver of informed consent for anonymous data collection was approved.

### 4.2. DNA Extraction

Genomic DNA was extracted following enzymatic lysis with lysozyme (Sigma, North Liberty, IA, USA), and a QIAamp DNA Mini kit (QIAGEN, Dusseldorf, Germany) was used. We prepared a bacterial suspension of 1.5 McFarland standard in 250 µL water containing 20 U of lysozyme, 2 mM EDTA (Protech, Taipei, Taiwan), 1.2% Triton (Sigma, North Liberty, IA, USA), and 20 mM Tris buffer. The DNA-containing supernatants were collected after incubation of the suspension at 56 °C for 30 min and 96 °C for 15 min. Repeated centrifugation of the lysates at 6000× *g* for 1 min was performed after buffer AW1 and AW2 (QIAGEN, Dusseldorf, Germany) were added.

### 4.3. CRISPR1 Locus Amplification

CRISPR1 locus amplification was performed in a TProfessionalTRIO thermocycler (Biometra, Gottingen, Germany) using CRISPR1 SEQ-F (GAAGACTCTATGATTTACCGC) and CRISPR1 SEQ-R (CAGCAATCACTAAAAGAACCAAC) primers targeting the CRISPR1-flanking regions, similar to previous studies [21]. PCR amplification was performed in a total volume of 25 μL, which contained 0.5 μM of each primer, 0.2 mM deoxynucleoside triphosphates (dNTPs), 0.02 U/μL HotStarTaq polymerase kit (QIAGEN, Dusseldorf, Germany), 1× PCR buffer, and 1 μL extracted DNA (0.05 μg/μL). The PCR mixtures were heated to 94 °C for 5 min, followed by 40 cycles of a denaturation step at 94 °C for 30 s, an annealing step at 58 °C for 30 s, an elongation step at 72 °C for 3 min, and a final extension step at 72 °C for 10 min. PCR amplification was verified by electrophoretic migration in a 1% agarose gel.

### 4.4. Restriction Fragment Length Polymorphism (RFLP) Analysis

We used NEBcutter V2.0 (http://nc2.neb.com/NEBcutter2/, accessed on 23 February 2021) to select the restriction enzymes according to previously sequenced CRISPR1 arrays. DdeI and MnlI were selected to test most of the CRISPR1 arrays. The RFLP analyses were performed on PCR products of the above CRISPR1 locus amplification, which contained the repeats and spacers. Restriction digests were performed in a total volume of 10 μL, containing 0.1 U/μL DdeI, 1× NEBuffer, and 4 μL extracted DNA (0.05 μg/μL) in a TProfessionalTRIO thermocycler (Biometra, Gottingen, Germany). Restriction products were analyzed by electrophoretic migration in a 1.5% agarose gel. The CRISPR1 locus was then sequenced in all CC17/III GBS isolates with different sizes and/or different patterns in RFLP.

### 4.5. CRISPR1 Locus Sequencing and Analysis

The DNA sequences of all CRISPR1 loci were analyzed and assembled using Sequencing analysis software v.5.3.1 (Applied Biosystems, Waltham, MA, USA). All the PCR products were sequenced in both the forward and reverse directions to obtain double-stranded sequences. All sequences of the spacers, repeats, and flanking regions were assembled using SeqMan software (DNASTAR, Madison, WS, USA) and then submitted to the CRISPR-finder database (https://crispr.i2bc.paris-saclay.fr/Server/, accessed 23 February 2021) (the spacer Id in the database: 2493099). Then spacer sequences were compared to CRISPRtionary (https://crispr.i2bc.paris-saclay.fr/CRISPRcompar/Dict/Dict.php, accessed 23 February 2021), and graphic representations of spacers were produced as colored cells in Excel spreadsheets. The CRISPR spacers were aligned in the CRISPR-finder database and named manually. New spacers identified in the GBS strains were assigned a unique number preceded by the letter “S”.

Based on the sequences of CRISPR spacers, BLASTn (default parameters, nr database, and short queries) was used to identify phage genes, plasmids, or GBS chromosome harboring sequences with homology to the spacers. The phage–host interactions and the evolutionary changes between phages and bacteria could be speculated on after examining the arrangement of spacers. The phylogenetic relationship of the GBS strains based on CRISPR diversity was constructed using Bionumeriucs 7.0 software (DNASTAR, Madison, WS, USA).

### 4.6. MLST and Capsule Genotyping

The MLST and capsule genotyping of the CC17/III GBS isolates were performed previously based on the standard protocol in previous studies [39].

### 4.7. Antimicrobial Susceptibility Testing

All GBS isolates were rated for susceptibility to seven antibiotics, including erythromycin, penicillin, clindamycin, vancomycin, ampicillin, cefotaxime, and teicoplanin, according to the guidelines of the Clinical and Laboratory Standards Institute for the microdilution minimum inhibitory concentration (MIC) method [40]. The double-disk diffusion test was applied to identify inducible clindamycin resistance.

### 4.8. Whole-Genome Sequencing (WGS)

WGS was conducted to investigate the genomic differences between CC17/III GBS isolates with different RFLP patterns and CRISPR1 locus. WGS was performed using both PacBio^TM^ SMRT (Pacific Biosciences, Menlo Park, CA, USA) [41] and MiSeq^TM^ (Illumina, San Diego, CA, USA) [42] sequencing technologies. The sequencing library was prepared using a TruSeq DNA LT Sample Prep Kit (Illumina, San Diego, CA, USA) for the Illumina MiSeq system. Genomic libraries were generated using Nextera XT kits (Illumina, San Diego, CA, USA). All the sequencing processes were performed using a DNA-sequencing kit 4.02v2 (QIAGEN, Dusseldorf, Germany) and SMRT cell 8 Pac (QIAGEN, Dusseldorf, Germany). The circlator tool (v1.4.0) (Cambridge, UK) was used to correct and linearize the genome, and QUAST (v4.5) (San Diego, CA, USA) was applied to evaluate the assembled genome quality.

## Figures and Tables

**Figure 1 ijms-22-11626-f001:**
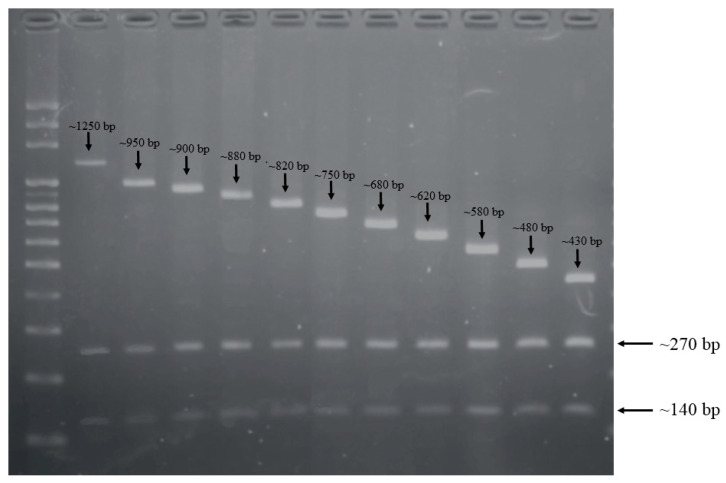
Restriction fragment length polymorphism (RFLP) results of CRISPR1 from 103 CC17/III GBS isolates. The length of CRISPR1 sequences after restriction enzyme digestions ranged from 140 to 1250 bp. The 140 and 270 bp fragments existed in all CC17/III GBS isolates, and the third fragment with spacers had lengths ranging from 430 to 1250 bp.

**Figure 2 ijms-22-11626-f002:**
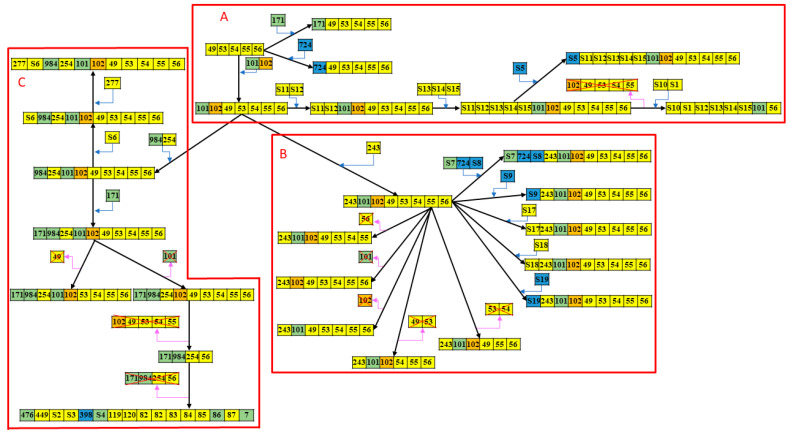
Phylogenetic relationship of 103 CRISPR-positive GBS strains constructed on 43 CRISPR types based on eBURST analysis. All 103 CC17/III GBS isolates can be categorized into three major subgroups (**A**–**C**) based on the CRISPR-RFLP array and eBURST analysis. The differences between compositions of CRISPR and gain or loss of spacers are marked and their relationships are presented.

**Table 1 ijms-22-11626-t001:** Correlation of the CRISPR-RFLP pattern with spacer compositions.

Length of CRISPR-RFLP (bp)	No. of Isolates (%)	Fragment Length (bp)			No. of Isolates (%)		CRISPR1 Spacers															
~1660	1 (1)	~1250	~270	~140	1	(1)	476	449	S2	S3	398	S4	119	120	82	82	83	84	85	86	87	7
~1360	2 (1.9)	~950	~270	~140	2	(1.9)				S5	S11	S12	S13	S14	S15	101	102	49	53	54	55	56
~1310	2 (1.9)	~900	~270	~140	2	(1.9)					S11	S12	S13	S14	S15	101	102	49	53	54	55	56
~1290	2 (1.9)	~880	~270	~140	1	(1)						277	S6	984	254	101	102	49	53	54	55	56
					1	(1)						S7	724	S8	243	101	102	49	53	54	55	56
~1230	23 (22.3)	~820	~270	~140	22	(21.4)							171	984	254	101	102	49	53	54	55	56
					1	(1)							S6	984	254	101	102	49	53	54	55	56
~1160	27 (26.2)	~750	~270	~140	14	(13.6)								984	254	101	102	49	53	54	55	56
					3	(2.9)								171	984	254	101	102	53	54	55	56
					3	(2.9)								S18	243	101	102	49	53	54	55	56
					2	(1.9)								S9	243	101	102	49	53	54	55	56
					2	(1.9)								S17	243	101	102	49	53	54	55	56
					1	(1)								171	984	254	102	49	53	54	55	56
					1	(1)								S19	243	101	102	49	53	54	55	56
					1	(1)								S11	S12	101	102	49	53	54	55	56
~1090	21 (20.4)	~680	~270	~140	20	(19.4)									243	101	102	49	53	54	55	56
					1	(1)									S10	S1	S12	S13	S14	S15	101	56
~1030	4 (3.9)	~620	~270	~140	1	(1)										243	101	102	49	53	54	55
					1	(1)										101	102	49	53	54	55	56
					1	(1)										243	102	49	53	54	55	56
					1	(1)										243	101	49	53	54	55	56
~990	5 (4.9)	~580	~270	~140	2	(1.9)											724	49	53	54	55	56
					1	(1)											243	101	102	49	55	56
					1	(1)											171	49	53	54	55	56
					1	(1)											243	101	102	54	55	56
~890	12 (11.7)	~480	~270	~140	12	(11.7)												49	53	54	55	56
~840	4 (3.9)	~430	~270	~140	4	(3.9)													171	984	254	56

Highlight in yellow, spacer (viral DNA, phage); orange, spacer (plasmid); green, spacer (chromosomal sequences); blue, spacer (unmatched).

**Table 2 ijms-22-11626-t002:** Clinical presentations of 103 cases of GBS invasive diseases categorized based on the length of the CRISPR-RFLP array and eBURST analysis.

Group	A	B	C
Case number, *n* (% of total)	22 (21.4)	34 (33.0)	47 (45.6)
RFLP patterns			
<1000 bp	15 (68.2)	2 (5.9)	4 (8.5)
1000–1300 bp	3 (13.6)	32 (94.1)	42 (89.4)
>1300 bp	4 (18.2)	0 (0)	1 (2.1)
Birth body weight (g)	2914.8 ± 430.2	2880.0 ± 644.5	3002.6 ± 683.0
Gestational age (weeks)	38.5 ± 1.8	37.6 ± 3.1	38.0 ± 3.1
Prematurity	3 (13.6)	8 (23.5)	8 (17.0)
Sex (male/female)	7 (31.8)/15 (68.2)	24 (70.6)/10 (29.4)	18 (38.3)/29 (61.7)
Any chronic comorbidity	1 (4.5)	3 (8.8)	5 (10.6)
Antibiotic susceptibility			
Erythromycin (R)	17 (77.3)	2 (82.4)	47 (100.0)
Clindamycin (R)	15 (68.2)	2 (70.6)	47 (100.0)
Clinical presentations			
Severe sepsis *	14 (63.6) **	7 (20.6)	18 (38.3)
Meningitis	11 (50.0) **	6 (17.6)	12 (25.5)
Neurological sequelae	6 (27.3)	4 (11.8)	6 (12.8)
Final mortality	1 (4.5)	1 (0)	2 (4.3)

All data are presented as number (%); R: resistance; RFLP: restriction fragment length polymorphism; * Severe sepsis is defined as the presence of septic shock, respiratory failure requiring intubation, and/or multiple organ failure; ** *p* < 0.05.

**Table 3 ijms-22-11626-t003:** Spacer sequences of all CRISPR loci in the 103 GBS isolates and the potential sources and function using BLASTN analysis.

Spacer	Sequence	Homology Analysis	Homology Percentage	Function
Viral DNA (phages), 65.1% (*n* = 28)				
49	TGCTAAAAGGTAAATTTAACATTCCAGGTA	Streptococcus phage LF2	100%	major tail protein
53	TATTTGATAGCGGTAACGGGTCATATACAA	Streptococcus phage Javan284	76%	putative C5 methylase
54	TGGTGGTATTTATAATGTACGAGCAAATCG	Streptococcus phage Javan52	100%	tail fibers protein
55	GATAAAAAGTGGGAGCTGAATTAAAAGGCA	Streptococcus phage Javan52	100%	hypothetical protein
56	ATTTGAACGATTTTTATATTCCTGATATGT	Streptococcus phage Javan516	100%	lysin, *N*-acetylmuramoyl-l-alanine amidase
82	CGTACCATCTATCAATTTACCGCAAGCTGT	Streptococcus phage LF2	100%	hypothetical protein
83	CCGATTATTTCCTACATAATACGCACGTTT	Streptococcus phage LF2	100%	hypothetical protein
84	AACTGTGTGATACTTTTCGTTTTTTTCTTT	Streptococcus phage Javan7	100%	hypothetical protein
85	TTTTATAAGTGATAGAGTGTGCAACACCGT	Streptococcus phage JX01	100%	putative minor structural protein
87	TTCTAAATGCTGGTGACTGCTTTGCATAAA	Streptococcus phage LF2	100%	hypothetical protein
119	GCGATGATGGTAAGTCATCATGGACAGCGT	Streptococcus phage Javan48	100%	tail fibers protein
120	TTTTACACACGATGTCAGATATAATGTCAA	Streptococcus phage Javan10	100%	membrane protein
243	TTGACCGCTCGTCCATTTTTTTAATGTAAA	Streptococcus phage Javan48	100%	tail fibers protein
254	ACCTTGCTCCGATGACACCATCGCGAACCT	Streptococcus phage Javan52	100%	tail fibers protein
277	AATTGATTGCCGTTAAAACCGATAGAGGA	Streptococcus phage LF2	100%	structural protein
449	TAAAATCCTGAAACAGAATGGGATTGATAT	Streptococcus phage Javan90	82%	antirepressor protein
S1	AATTGATACATTGCAACGTCTAGCAGGAGC	Streptococcus phage LF2	100%	tail length tape-measure protein
S2	GTGTGTTCTTCATTTTTATCAAACCAAAA	Streptococcus phage Javan471	100%	hypothetical protein
S3	AAGAAATTCGGTAGAGACCCCAGACTCAT	Streptococcus phage Javan46	100%	tail length tape-measure protein
S6	ATTAAATCTTCTTTTGAAGTTACTGTACGT	Staphylococcus phage pSco-10	70%	hypothetical protein
S10	TTTATATTGTTCAGAAGAATGCCGCAAAAA	Streptococcus phage Javan44	100%	Phage-associated protein
S11	TGTGTACGTTGCCTTTCCGTCAGCACCAGC	Streptococcus phage Javan52	100%	tail fibers protein
S12	CCATAAACTTGCCAGTAGATGTGTCACGCT	Streptococcus phage Javan648	100%	hypothetical protein
S13	ACCATTCGAAGTAGCTAGTTTGATTTCGTA	Streptococcus phage LF2	100%	tail fibers protein
S14	TGTCGATGGTGTTCAAATACAAATGTTTTC	Streptococcus phage Javan52	100%	hypothetical protein
S15	CTTTACCATTATTGATTTGTTCTTGCTTTT	Streptococcus phage Javan478	100%	hypothetical protein
S17	TCGCAATAATTACTATATGCTTAAGCGGAG	Streptococcus phage Javan7	96%	Phage protein
S18	AATCCAACAAAAACAACTTGCTTTAAATAA	Streptococcus phage Javan48	100%	membrane protein
Plasmid, 2.3% (*n* = 1)				
102	CTGTTCATAAAGAGCAACTAGTGGCAACAT	Bacillus megaterium strain YC4-R4 plasmid unnamed2	70%	hypothetical protein
Chromosomal sequences, 18.6% (*n* = 8)				
7, 86, 101, 171, 476, 984, S4, S7		GBS chromosome	x	x
Unmatched, 14.0% (*n* = 6)				
398, 724, S5, S8, S9, S19		Unmatch	x	x

## Data Availability

The datasets used/or analyzed during the current study are available from the corresponding author on reasonable request.

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
