# Peer review of "Clustered Regularly Interspaced Short Palindromic Repeat Analysis of Clonal Complex 17 Serotype III Group B Streptococcus Strains Causing Neonatal Invasive Diseases"

_ijms, 2021, doi:10.3390/ijms222111626_

Round 1
Reviewer 1 Report
I reviewed the improved version of the work of Hsu et. al. entitled "Clustered regularly interspaced short palindromic repeat analysis of clonal complex 17 serotype III group B Streptococcus strains causing neonatal invasive diseases". After revision I accept all changes and improvement of the paper. In my opinion the work should be published in the International Journal of Molecular Sciences.
One minor comment:
Add name of manufacturer of buffers AW1 and AW2 (Qiagen – as you wrote in answers to my comments).
Author Response
Comments from Reviewer No.1
I reviewed the improved version of the work of Hsu et. al. entitled “Clustered regularly interspaced short palindromic repeat analysis of clonal complex 17 serotype III group B Streptococcus strains causing neonatal invasive diseases”. After revision I accept all changes and improvement of the paper. In my opinion the work should be published in the International Journal of Molecular Sciences.
One minor comment:
Add name of manufacturer of buffers AW1 and AW2 (Qiagen – as you wrote in answers to my comments)
Reply:
Thank you for your appreciated comments. I will add the name “Qiagen” in line 97 according, thank you.
Reviewer 2 Report
Authors of the manuscript addressed most of the issues, reported previously. They provided text modifications according to their scientific view on the problem.
Authors provided explanation how to find their spacers submitted to the database in their response. However they didn't include it in the main text. They provided database id: 2493099 which can be used to identify their spacers (I can't check it, since technical problems on https://crispr.i2bc.paris-saclay.fr/ side). They should include this Id in text or supplement, so readers could easily find the spacer sequences.
I think that the text in acceptable shape to be published, when following issues will be corrected:
Line 138-139: language must be corrected
Line 188: "always" is incorrect, it happens in most of the cases, but not always
Author Response
Comments from Reviewer No.2:
Authors of the manuscript addressed most of the issues, reported previously. They provided text modifications according to their scientific view on the problem.
Authors provided explanation how to find their spacers submitted to the database in their response. However they didn’t include it in the main text. They provided database id: 2493099 which can be used to identify their spacers (I can’t check it, since technical problems on https://crispr.i2bc.paris-saclay.fr/side). They should include this ID in the text or supplement, so readers could easily find the spacer sequences.
Reply:
Thank you for your appreciated comments. I will provide this Id in the text (in line 126: (the spacer Id in the database: 2493099)).
I think that the text in acceptance shape to be published, when following issues will be corrected:
Line 138-139: language must be corrected
Reply:
Thank you for your instructive advice. I correct this sentence as the following:
“The MLST and capsule genotyping of the CC17/III GBS isolates had been performed previously based on the standard protocol in previous studies”
Line 188: “always” is incorrect, it happens in most of the cases, but not always.
Reply:
Thank you for your appreciated comments. I correct this sentence as “…..are inserted at 5’ end of the array in most of the cases,” accordingly, thank you.
This manuscript is a resubmission of an earlier submission. The following is a list of the peer review reports and author responses from that submission.
Round 1
Reviewer 1 Report
The work of Hsu et. al. entitled "Clustered regularly interspaced short palindromic repeat analysis of clonal complex 17 serotype III group B Streptococcus strains causing neonatal invasive diseases" describes an in silico analysis of the CRISPR1 region in 103 Streptococcus isolates. In my opinion, that's a purely descriptive work which is a good starting point for performing molecular work in the future. I cannot agree with the conclusions the authors drew in the paper e.g.: i) “Therefore, we found that The CC17/III GBS isolates with RFLP lengths ≥ 1300 bp and < 1000 bp were significantly more likely to cause severe sepsis (57.1% vs. 29.3%, P=0.012) and meningitis (50.0% vs. 20.8%, P=0.009) than GBS isolates with RFLP lengths between 1000 bp and 1300 bp” or “we investigated the in vivo dynamics of invasive GBS strains by comparing the CRISPR1 loci of 103 CC17/III GBS isolates from 103 different neonates over an 11-year period”. In order to come to such conclusions, the authors would have to perform a molecular analysis of the mentioned CRISPR1 regions.
The strong point of this work is a description of a CRISPR-RFLP technique as a useful tool for the identification of CC17/III GBS isolates. However, the description on the Materials & Methods section should be improved.
Comments and questions:
- Lines 80-82: “All identifications of initial GBS isolates and storage in the bacterial library were based on the standard manufacturer’s recommendation” – who is the manufacturer? Where should the readers look for such recommendations?
- Lines 93-94: “Repeated centrifugation of the lysates at 6,000 × g for 1 minute was done after buffer AW1 and AW2 were added.” – add composition of the buffers.
- Lines 103-104: “…(dNTPs), 0.02 U/µl HotStarTaq polymerase kit (QIAGEN), 1x PCR buffer, and 1 µl extracted DNA.” – add the concentration of extracted DNA.
- Line 112: “…volume of 10 µl, containing 0.1 U/µl DdeI, 1x NEBuffer, and 4 µl extracted DNA…” – add the concentration of extracted DNA.
- Materials & Methods 2.6 section – there are no MLST results in the paper. This section should be deleted.
- Figure 1. I am confused. In the manuscript text, the authors refer to this figure as follows “The resulting restriction fragments were separated according to length by gel electrophoresis (Figure 1).” However, the figure captions says: “PCR results of CRISPR1 in 103 CC17/III GBS isolates”. So, please decide if it is the visualization of PCR fragments or restriction digestion results.
- 1. – did the authors carry out a detailed analysis of 270-bp and 140-bp fragments from all 11 pattern variants? It would be interesting.
- Lines 174-175: “A total of 18 new spacers were found in this cohort, submitted to GeneBank and assigned accession numbers S1 to S18.” – S1 -S18 are not the correct GenBank accession numbers..
- Lines 203-206: “Therefore, we found that The CC17/III GBS isolates with RFLP lengths ≥ 1300 bp and < 1000 bp were significantly more likely to cause severe sepsis (57.1% vs. 29.3%, P=0.012) and meningitis (50.0% vs. 20.8%, P=0.009) than GBS isolates with RFLP lengths between 1000 bp and 1300 bp.” – it is only an observation. To conclude that such RFLP length (≥ 1300 bp and < 1000 bp) causes sepsis or meningitis more frequently the authors have to prove it experimentally (not only as an observation of one chosen DNA fragment). I am sure more factors are implicated in the process.
- Figure 2. – add more detailed description in figure caption. In this form I really do not understand this figure and probably no one except the authors understand it.
- Lines 239-241: “In this study, we investigated the in vivo dynamics of invasive GBS strains by comparing the CRISPR1 loci of 103 CC17/III GBS isolates from 103 different neonates over an 11-year period.” – please add the results of such an analysis. I did not find them in the paper.
Author Response
Comments from Reviewer No.1:
The work of Hsu et. al. entitled "Clustered regularly interspaced short palindromic repeat analysis of clonal complex 17 serotype III group B Streptococcus strains causing neonatal invasive diseases" describes an in silico analysis of the CRISPR1 region in 103 Streptococcus isolates. In my opinion, that's a purely descriptive work which is a good starting point for performing molecular work in the future. I cannot agree with the conclusions the authors drew in the paper e.g.: i) “Therefore, we found that The CC17/III GBS isolates with RFLP lengths ≥ 1300 bp and < 1000 bp were significantly more likely to cause severe sepsis (57.1% vs. 29.3%, P=0.012) and meningitis (50.0% vs. 20.8%, P=0.009) than GBS isolates with RFLP lengths between 1000 bp and 1300 bp” or “we investigated the in vivo dynamics of invasive GBS strains by comparing the CRISPR1 loci of 103 CC17/III GBS isolates from 103 different neonates over an 11-year period”. In order to come to such conclusions, the authors would have to perform a molecular analysis of the mentioned CRISPR1 regions.
The strong point of this work is a description of a CRISPR-RFLP technique as a useful tool for the identification of CC17/III GBS isolates. However, the description on the Materials & Methods section should be improved.
Reply:
Thank you for your appreciated comments. I will describe that “we found that the CC17/III GBS isolates with RFLP lengths ≥ 1300 bp and < 1000 bp were significantly associated with the occurrences of severe sepsis and meningitis……” as the observational result instead of the conclusion. Therefore, in the abstract, the conclusion will be “The CRISPR-RFLP analysis is potentially applicable to categorize CC17/III GBS isolates, and a specific CRISPR-RFLP pattern could be used as a new biomarker to predict meningitis and illness severity after further verification.”
For the “we investigated the in vivo dynamics of invasive GBS strains by comparing comparing the CRISPR1 loci of 103 CC17/III GBS isolates from 103 different neonates over an 11-year period” (in the first sentence of discussion), I will revise it as “the molecular characteristics of…….” (line 241)
Comments and questions:
1. Lines 80-82: “All identifications of initial GBS isolates and storage in the bacterial library were based on the standard manufacturer’s recommendation” – who is the manufacturer? Where should the readers look for such recommendations?
Reply:
Thank you for your instructive advice. I think my description will case misunderstanding, and I am sorry about that. I will revise as “All identifications of initial GBS isolates and storage in the bacterial library were based on the standard procedures: the MALDI-TOF MS (Bruker Daltonics Germany) was used for documentation of all GBS isolates (https://www.ncbi.nlm.nih.gov/pmc/articles /PMC8328194/). (line 80-82)
2. Lines 93-94: “Repeated centrifugation of the lysates at 6,000 × g for 1 minute was done after buffer AW1 and AW2 were added.” – add composition of the buffers.
Reply:
Thank you for your instructive advice. The buffer AW1 and AW2 are the reagents with the QIAamp DNA Mini kit (QIAGEN). We have contacted the QIAGEN manufacturer for the composition of the buffers. However, the representative informed us that the compositions of AW1 and AW2 are business secrets and they cannot tell us. Therefore, I am afraid that I cannot provide this information, sorry.
3. Lines 103-104: “…(dNTPs), 0.02 U/µl HotStarTaq polymerase kit (QIAGEN), 1x PCR buffer, and 1 µl extracted DNA.” – add the concentration of extracted DNA.
Reply:
Thank you for your instructive advice. The concentration of extracted DNA was 0.05 μg/μl. I will add the concentration of extracted DNA in the revised manuscript, lines 104
4. Line 112: “…volume of 10 µl, containing 0.1 U/µl DdeI, 1x NEBuffer, and 4 µl extracted DNA…” – add the concentration of extracted DNA.
Reply:
Thank you for your instructive advice. The concentration of extracted DNA was (0.05 μg/μl). I will add the concentration of extracted DNA in the revised manuscript, line 114
5. Materials & Methods 2.6 section – there are no MLST results in the paper. This section should be deleted.
Reply:
Thank you for your instructive advice. I delete the 2.6 section regarding the methods of MLST and capsular analyses. I just mentioned that we have done the MLST and serotyping in line 136-137
6. Figure 1. I am confused. In the manuscript text, the authors refer to this figure as follows “The resulting restriction fragments were separated according to length by gel electrophoresis (Figure 1).” However, the figure captions says: “PCR results of CRISPR1 in 103 CC17/III GBS isolates”. So, please decide if it is the visualization of PCR fragments or restriction digestion results.
Reply:
Thank you for your instructive advice. The figure 1 is the restriction digestion results. I will revise the figure 1 legend as “Restriction fragment length polymorphism (RFLP) results of CRISPR1 from 103 CC17/III GBS isolates. The length of CRISPR1 sequences after restriction enzyme digestions ranged from 140 bp to 1250 bp. The 140-bp and 270-bp fragments existed in all CC17/III GBS isolates, and the third fragment with spacers had lengths ranged from 430 bp to 1250 bp.” (L177-180)
7. 1. – did the authors carry out a detailed analysis of 270-bp and 140-bp fragments from all 11 pattern variants? It would be interesting.
Reply:
Thank you for your instructive advice. We analyzed the 140-bp and 270-bp fragments using NCBI for Blast analysis. We found that the 140-bp fragment is part of the type II-A crispr associated protein and the 270-bp fragment is the hypothetical protein without specific function. All CC17/III GBS isolates had these two fragments. I add a result description “The common 140-bp and 270-bp fragments were part of the type II-A crispr associated protein and hypothetical protein, respectively.” in the 2nd paragraph of the result section (line 160-161)
8. Lines 174-175: “A total of 18 new spacers were found in this cohort, submitted to GeneBank and assigned accession numbers S1 to S18.” – S1 -S18 are not the correct GenBank accession numbers.
Reply:
Thank you for your instructive advice. We have submitted the 18 new spacers to GeneBank in recent days and the application number is 2493099. Of course the numbers S1 to S18 are not the correct GenBank accession numbers. Actually we assigned these new spacer numbers as S1, S2, S3, until S18 because we cannot find these new spacers in the CRISPRtionary (https://crispr.i2bc.paris-saclay.fr/CRISPRcompar/Dict/Dict.php). For convenience of description in this manuscript, we just assigned these new spacer numbers, thank you. I have mentioned in line 127-128 that this number was assigned by us. Therefore, in line 171, we will revise as “assigned as S1 to S18.”
9. Lines 203-206: “Therefore, we found that The CC17/III GBS isolates with RFLP lengths ≥ 1300 bp and < 1000 bp were significantly more likely to cause severe sepsis (57.1% vs. 29.3%, P=0.012) and meningitis (50.0% vs. 20.8%, P=0.009) than GBS isolates with RFLP lengths between 1000 bp and 1300 bp.” – it is only an observation. To conclude that such RFLP length (≥ 1300 bp and < 1000 bp) causes sepsis or meningitis more frequently the authors have to prove it experimentally (not only as an observation of one chosen DNA fragment). I am sure more factors are implicated in the process.
Reply:
Thank you for your instructive advice. I completely agree with your comments that this is only an observation. Therefore, I will revise it as “we found that the CC17/III GBS isolates with RFLP lengths ≥ 1300 bp and < 1000 bp were significantly associated with occurrences of severe sepsis (57.1% vs. 29.3%, P=0.012) and meningitis (50.0% vs. 20.8%, P=0.009) than GBS isolates with RFLP lengths between 1000 bp and 1300bp.” (line 209-210). As also in the abstract, this is a description of an observation. The final conclusion will be “The CRISPR-RFLP analysis is potentially applicable to categorize CC17/III GBS isolates, and a specific CRISPR-RFLP pattern could be used as a new biomarker to predict meningitis and illness severity after further verification.” (The last two sentences of the abstract).
10. Figure 2. – add more detailed description in figure caption. In this form I really do not understand this figure and probably no one except the authors understand it.
Reply:
Thank you for your instructive advice. I will add more detailed description in the figure 2 legend as the following: All 103 CC17/III GBS isolates can be categorized into three major subgroups (A, B, and C) based on the CRISPR-RFLP array and eBURST analysis. The differences between compositions of CRISPR and gain or loss of spacers are marked and their relationships are presented. (line 210-212)
11. Lines 239-241: “In this study, we investigated the in vivo dynamics of invasive GBS strains by comparing the CRISPR1 loci of 103 CC17/III GBS isolates from 103 different neonates over an 11-year period.” – please add the results of such an analysis. I did not find them in the paper.
Reply:
Thank you for your instructive advice. All the CRISPR1 loci analyses of 103 CC17/III GBS isolates from 103 different neonates over an 11-year period have been presented in the result section, including the sequencing data, their phylogenetic relationships (figure 2), the spacer compositions, and clinical features (Table 2). I think the question will be the “in vivo dynamics”. I will revise it as “In this study, we investigated the molecular characteristics of invasive GBS strains by comparing………” (line 241), thank you.
Reviewer 2 Report
Jen-Fu Hsu and Jang-Jih Lu et al. in the manuscript “Clustered Regularly Interspaced Short Palindromic Repeat Analysis of Clonal Complex 17 Serotype III Group B Streptococcus Strains Causing Neonatal Invasive Diseases'' study CRISPR1 loci of 103 Group B Streptococcus isolates using RFLP and other analyses. They split this set into three groups to show the differential relation of these groups with patient conditions. They try to assess the relation of RFLP to clinical manifestation.
The main strength of this work is the access to 103 Group B Streptococcus isolates from patients with diseases. Neonatal diseases and their relationship to CRISPR content of Streptococcus strains are an important topic for research which have been covered in a number of recent publications and this paper can be a good addition to this set. However a number of improvements can be done to make this manuscript more scientifically sound.
1: Authors must provide spacer Ids they submitted to the database, or the clear way to identify these spacers. Additionally it would be very helpful to have a table that will show correspondence of spacer Id and patient condition (unless it is forbidden formally).
2: Authors have RFLP, spacer sequences, related three groups and patient conditions. Using this information they assess significance of each of the groups to a certain state of a patient of Streptococcus strain. However it is important to have a deeper analysis in this case. Figure 3 shows the spacers that discriminate each of the three groups. Authors already have possible targets for these spacers. These targets should be described with more details and their possible effect should be discussed. I.e. if spacer targets a plasmid with resistance genes or virulence factors. So more insight could be given, to explain why these groups have different effects.
3: Authors should argue more regarding spacer acquisition and loss rate since it is important for their results. They have data on spacers in the CRISPR arrays for their isolates and other related strains of Streptococcus, so they could assess these rates to strengthen or weaken their results. Additionally the recent paper: “Functional Study of the Type II-A CRISPR-Cas System of Streptococcus agalactiae Hypervirulent Strains” by Adeline Pastuszka et al. can be used and challenged.
Additional comments:
Title: CRISPR is a well known acronym, so the title can be significantly shortened.
Line 59-63: Gene editing information is not relevant to this paper and can be omitted.
Line 108: Restriction fragment length polymorphism (RFLP) analysis. Text must contain clear information which regions were chosen for the restriction enzymes. For example, it might be not clear if repeats or regions adjacent for CRISPR1 were chosen.
Line 168: “Our data showed that a higher molecular weight CRISPR-RFLP band was significantly correlated with more spacers of the CRISPR1 loci.” It is trivial information that can be omitted. Additionally no significance or correlation described in the text.
Line 177: “which suggested that these spacers may have been acquired recently” This information should be verified by checking for the position of the spacers. New spacers in most cases are inserted in 5’ end of the array. Authors have this information, since they have sequencing data.
Line 186-192: These stains and their repeats are well studied, so there is no novelty in this part, unless authors see difference from canonic repeat sequences of these strains
Line 228: “This result indicated that horizontal gene transfer by plasmids was uncommon in CC17/III GBS strains.” It is not explained why or not properly supported by the presented data.
Author Response
Dear reviewer, please see the attachment, thank you.
Best regard,
Tsai Ming Horng
